# A Highly Active NiMoAl Catalyst Prepared by a Solvothermal Method for the Hydrogenation of Methyl Acrylate

Taolue Sun [1,2], Gang Wang [2,3], Xinpeng Guo [4], Zengxi Li [3], Erqiang Wang [1,2,3,*] and Chunshan Li [2,*]

1 Sino-Danish College, University of Chinese Academy of Sciences, Beijing 100049, China
2 Beijing Key Laboratory of Ionic Liquids Clean Process, Key Laboratory of Multiphase Complex Systems, Institute of Process Engineering, Innovation Academy for Green Manufacture, Chinese Academy of Sciences, Beijing 100190, China
3 School of Chemical Engineering, University of Chinese Academy of Sciences, Beijing 100049, China
4 School of Chemical Engineering, Xiangtan University, Xiangtan 411105, China
* Correspondence: wangerqiang@ucas.ac.cn (E.W.); csli@ipe.ac.cn (C.L.)

**Abstract:** In this study, a series of $Ni_{10}Mo_xAl$ composite metal oxide ($Ni_{10}Mo_xAl$, NiO = 10 wt.%, x = 2.5, 5, 10, 15, 20 wt.%) catalysts with different Mo content were prepared by a solvothermal method using a water—ethanol system. By employing various characterization technologies, it was confirmed that the suitable amount of the Mo element can not only promote the dispersion of the Ni species but also inhibit the formation of the inactive $NiAl_2O_4$ phase. Consequently, the hydrogenation activity of the $Ni_{10}Mo_xAl$ catalysts was affected by the particle size of the active components and the amount of the $NiAl_2O_4$ phase. As a result, the $Ni_{10}Mo_{10}Al$ catalyst showed the best catalytic performance on methyl acrylate hydrogenation, and the yield of methyl propionate can be increased from 53.7% to 89.5% at 100 °C and 1 MP $H_2$, compared with the $Ni_{10}Mo_{10}/\gamma–Al_2O_3$ catalyst prepared by a traditional impregnation method. The stability of the $Ni_{10}Mo_{10}Al$ catalyst was also investigated, and the catalyst can run stably for 23 h. The novel strategy adopted in this article provides a new direction for the preparation of high activity Ni–Mo catalysts.

**Keywords:** $Ni_{10}Mo_xAl$ composite metal oxide; solvothermal method; methyl acrylate; hydrogenation; Ni species; $NiAl_2O_4$ phase

## 1. Introduction

As an important industrial feedstock, methyl propionate (MP) is widely used in the production of nitro spray paint and varnish. In addition, MP is also an essential intermediate for pharmaceutical, pesticide and perfume products [1–3]. In recent years, the hydrogenation of methyl acrylate (MA) to MP has received increasing attention due to its high atomic utilization and simple production process. However, MA is easy to polymerize during hydrogenation at relatively high temperatures, and the reported catalysts are mainly focused on precious metal catalysts, which severely restrict industrial application [4–8]. Therefore, the development of non-noble metal catalysts with high hydrogenation activity under mild reaction conditions is significantly required.

At present, the catalysts reported for the hydrogenation of MA to MP are precious metal catalysts due to their high activity under mild conditions. Trzeciak et al. reported that the yield of MP could reach 100% with catalysis of rhodium triphenyl phosphite in a batch reaction system [7]. Shiraishi et al. found that the platinum nanocluster protected by poy(*N*-vinyl-2-pyrrolidone) exhibited excellent behavior in the hydrogenation of MA, which was affected by the metal particle size [8]. Chen et al. synthesized a kind of Pd(II) complex using polyfunctional phosphine ligands for MA hydrogenation under atmospheric pressure, and the conversion achieved was higher than 92% [9]. Chen et al. obtained 100% yield of MP from MA hydrogenation over a Pd catalyst [10]. However, the high cost and low storage of precious metals limit their application in industrial production. Ni-based

catalysts have gained much more attention for their comparable hydrogenation activity with precious metal catalysts and relatively low cost of preparation [11,12].

It is well known that the dispersion and particle size of active species in catalysts are crucial to the activity [13,14]. For the Ni-based catalyst, the Ni species will easily agglomerate and sinter at high temperatures during calcination or reduction. Therefore, considerable efforts have been made to solve this problem. Zhang et al. discovered that the introduction of the Mo element could improve the dispersion of the Ni species on $SiO_2$ support, and the catalyst shows relatively high activity with a Mo/Ni ratio of 0.1 [15]. Kordouli et al. prepared a Ni–Mo/$\gamma$–$Al_2O_3$ catalyst with hierarchical NiO flake-flower architectures by the coprecipitation method and found that the addition of Mo could decrease the size of nickel nanoparticles and inhibit the formation of catalytically inactive nickel aluminate. In addition, flake-flower architectures formed under a high content amount of nickel (49–52 wt.%) when the $MoO_3$ content was below 7 wt.%, and the Mo-Ni/ASA catalyst displayed a two times higher yield than a commercial NiMoP/$Al_2O_3$ due to the higher Ni dispersion [16]. Lv et al. demonstrated that the Ni–Mo/$\gamma$–$Al_2O_3$ catalyst prepared by the thermal decomposition of the layered double hydroxides Ni–Al–$[C_6H_4(COO)_2]^{2-}$–LDHs/$\gamma$–$Al_2O_3$ showed a better dispersion of the NiO and $MoO_3$ on the alumina surface, and the yield of the catalyst increased by 20% more than the catalyst prepared by an impregnation method, while the preparation process was complicated and required a long time [17]. Past research has demonstrated that the dispersion of the active species can be improved by optimizing the catalyst preparation method. However, the coprecipitation method and the hydrothermal method limited the content of the active components in the catalyst due to the fact that Ni ions precipitate in alkaline environments, and Mo ions precipitate in acidic conditions. According to our previous work, the Ni–Mo/$\gamma$–$Al_2O_3$ catalyst, prepared by the incipient wetness impregnation method, exhibited excellent activity in the MA hydrogenation to MP under mild conditions; however, it still needs to be improved for industrial application [18]. Stimulated by the effects of dispersion and the particle size of the active species on catalytic activity, solvothermal technology was employed for the modification of this Ni–Mo–Al catalyst.

In this study, the $Ni_{10}Mo_xAl$ composite metal oxide ($Ni_{10}Mo_xAl$) catalysts with different Mo content (x = 2.5, 5, 10, 15, 20 wt.%) were prepared by a solvothermal method using a water–ethanol system. The physicochemical properties of the catalyst series were characterized using X-ray diffraction (XRD), Brunauer–Emmett–Teller analysis (BET), UV-vis diffuse reflectance spectroscopy (UV-vis DRS), hydrogen temperature programmed reduction ($H_2$-TPR), X-ray photoelectron spectroscopy (XPS) and transmission electron microscopy (TEM). The relationship between the structure and catalytic hydrogenation activity in these Ni–Mo–Al composite oxides were analyzed. In addition, the effects of the reaction conditions were systematically investigated.

## 2. Results and Discussion

### 2.1. Characterization of Catalysts

SEM analysis was carried out to study the morphological structure of the $Ni_{10}Mo_xAl$ catalyst precursors with different Mo content. For comparison, the SEM images of the $Ni_{10}Al$ and $Mo_{10}Al$ catalyst precursors were also obtained, as shown in Figure S1. Only small sheet structures were observed on the surface of the $Ni_{10}Al$, while large block solids were found on the surface of the $Mo_{10}Al$. For the $Ni_{10}Mo_xAl$ catalyst precursors, as shown in Figure 1, the $Ni(OH)_2$ nano-flower architectures were observed; this is dependent on the content of molybdenum [19,20]. The formation and mechanistic route of the $Ni(OH)_2$ nano-flower architectures and the maintenance of this structure after calcination (the transformation of $Ni(OH)_2$ to NiO phase) has been reported [21–23]. It was also demonstrated that this structure promotes the diffusion and hydrogenation of the reactant. When the Mo content was 2.5 wt.% (Figure 1a), an outward diffusion sheet structure was seen on the surface. With the increase in Mo content, a hierarchical structure gradually formed on the surface, as shown in Figure 1b. The hierarchical structure was observed more obviously

with a further increase in Mo content (Figure 1c), indicating that the introduction of the Mo element can promote the formation of hierarchical structures. However, when the Mo content attained 15 wt.% (Figure 1d), the hierarchical structure was covered by block solids. This phenomenon was more seriously observed with a further increase in Mo content to 20 wt.%, as shown in Figure 1e. In addition, the energy dispersive X-ray spectroscopy (EDS) elemental mapping of the $Ni_{10}Mo_{10}Al$ catalyst precursor was also conducted. As shown in Figure 1f, the Mo and Ni displayed a uniform distribution on the surface [24]. In addition, the content of the Ni and Mo tested using EDS was 6.55 wt.% and 6.35 wt.%, respectively, which were consistent with the ICP–OES analysis, as presented in Table 1. In addition, it can be seen from Table 1 that the content of NiO was close to the theory design value, and $MoO_3$ was also close to the theoretical design value when the content was below 15 wt.%.

**Table 1.** ICP–OES results for the fresh $Ni_{10}Mo_xAl$ catalysts.

| Catalyst | Ni Content [a] (wt%) | Mo Content [a] (wt%) |
|---|---|---|
| $Ni_{10}Mo_{2.5}Al$ | 6.85 | 1.29 |
| $Ni_{10}Mo_5Al$ | 6.59 | 2.90 |
| $Ni_{10}Mo_{10}Al$ | 6.58 | 6.07 |
| $Ni_{10}Mo_{15}Al$ | 6.55 | 8.42 |
| $Ni_{10}Mo_{20}Al$ | 6.59 | 11.15 |

[a] Content of Ni and Mo calculated on the ICP-OES analysis.

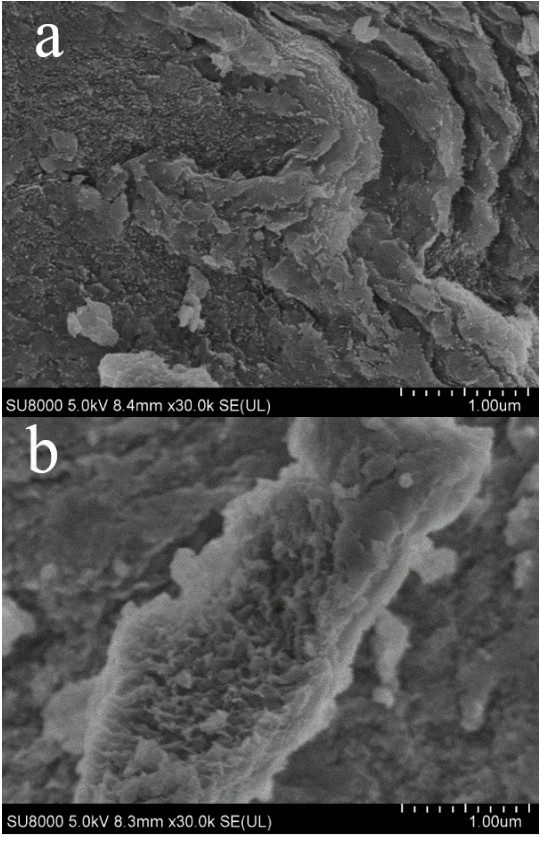

**Figure 1.** *Cont.*

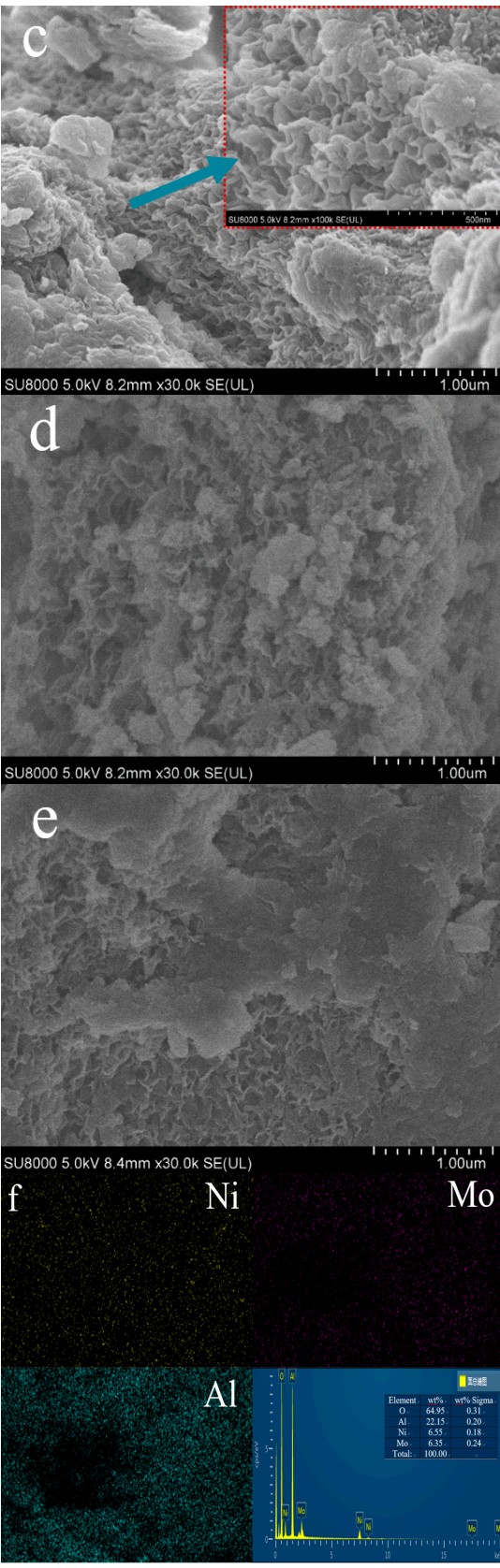

**Figure 1.** SEM images of (**a**) $Ni_{10}Mo_{2.5}Al$, (**b**) $Ni_{10}Mo_5Al$, (**c**) $Ni_{10}Mo_{10}Al$, (**d**) $Ni_{10}Mo_{15}Al$ and (**e**) $Ni_{10}Mo_{20}Al$ catalyst precursor; and (**f**) EDS mapping of Ni, Mo, Al elements for $Ni_{10}Mo_{10}Al$ catalyst precursor.

Figure 2 shows the TEM images of the reduced $Ni_{10}Mo_xAl$ catalyst series and the HRTEM image of the reduced $Ni_{10}Mo_{10}Al$ catalyst. It was noticed that the Ni species in the catalyst showed a smaller particle size with the increase in $MoO_3$ content in the region of 2.5–10 wt.%, suggesting that the introduction of Mo promotes the dispersion of the Ni species. A previous study also demonstrated that the introduction of Mo onto a Ni-based catalyst can inhibit the aggregation of the Ni species [25]. However, an excessive amount of the Mo element (15 wt.% and 20 wt.%) results in the growth of particle size because of the coverage of the Ni species by the excess Mo element, which was also demonstrated by the SEM results [26]. Figure 2f shows the HRTEM image of the $Ni_{10}Mo_{10}Al$ catalyst. An Ni particle with a lattice spacing of 0.204 nm was found, which corresponded to the plane of Ni (111). For comparison, a TEM image of $Ni_{10}Mo_{10}/\gamma-Al_2O_3$ catalyst was also obtained, as shown in Figure S2. The aggregation of the Ni species with a wide range of particle distribution ranging from 1 to 11 nm was clearly observed. In addition, the mean particle size was twice as large as that of the $Ni_{10}Mo_{10}Al$ catalyst, indicating that the solvothermal method is effective for the dispersion of the active components in the catalyst.

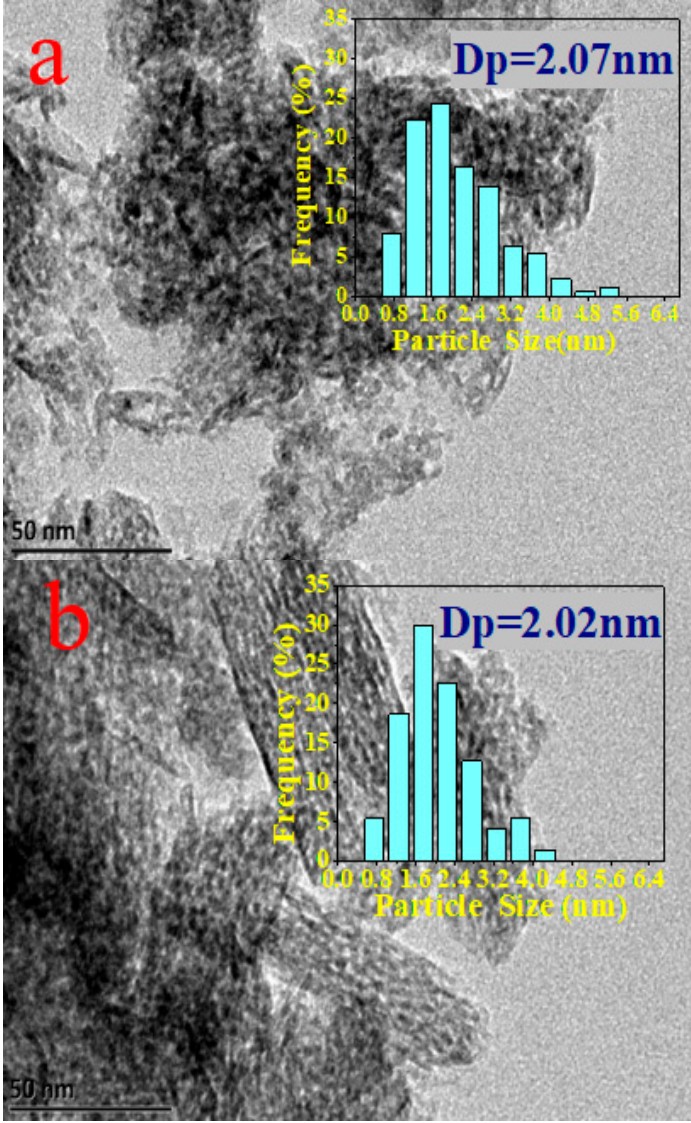

**Figure 2.** *Cont.*

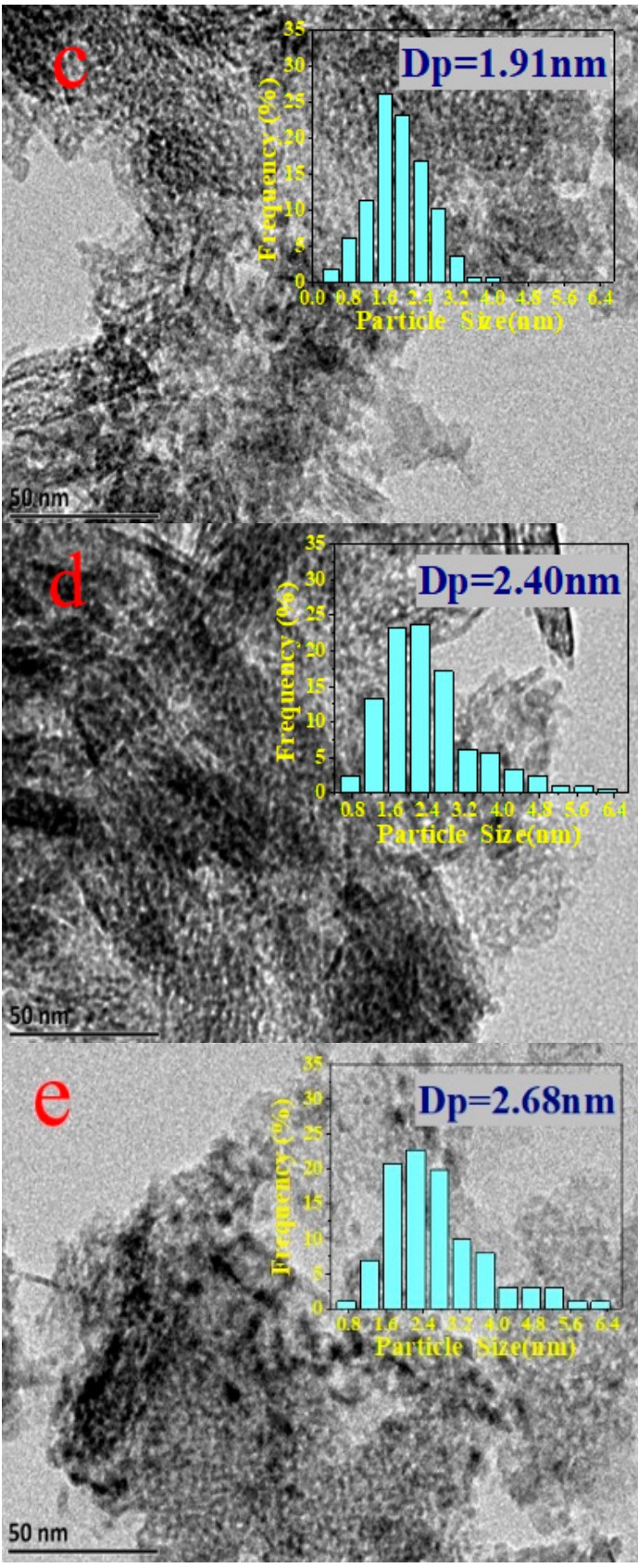

**Figure 2.** *Cont.*

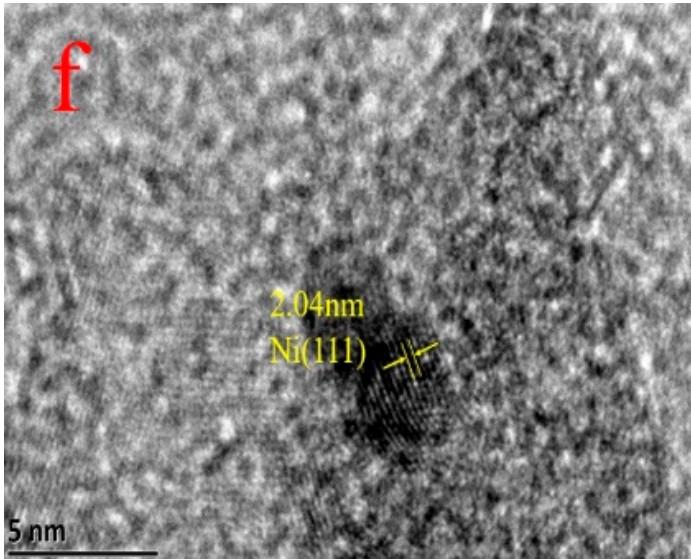

**Figure 2.** TEM images of (**a**) Ni$_{10}$Mo$_{2.5}$Al, (**b**) Ni$_{10}$Mo$_5$Al, (**c**) Ni$_{10}$Mo$_{10}$Al, (**d**) Ni$_{10}$Mo$_{15}$Al and (**e**) Ni$_{10}$Mo$_{20}$Al; and (**f**) HRTEM image of Ni$_{10}$Mo$_{10}$Al catalyst after reduction at 450 °C.

Figure 3 shows the TEM images of the reduced Ni$_{10}$Mo$_{10}$Al catalysts obtained under different calcination temperatures (450–750 °C). It was observed that the agglomeration of the active components, together with a wider distribution in particle size, became more obvious with the enhancement of the calcination temperature, indicating that a higher calcination temperature is unfavorable for the distribution of active components.

Figure 4 shows the N$_2$ adsorption–desorption isotherms and pore size distribution curves of the Ni$_{10}$Mo$_x$Al catalysts. All the catalysts showed the typical IV N$_2$ adsorption–desorption isotherms, as shown in Figure 4a, indicating that the mesoporous structure existed in these catalysts [27]. The pore size distribution curves, as shown in Figure 4b, also identified the mesoporous structure in these catalysts, and the average pore size gradually decreased with the increase in Mo content.

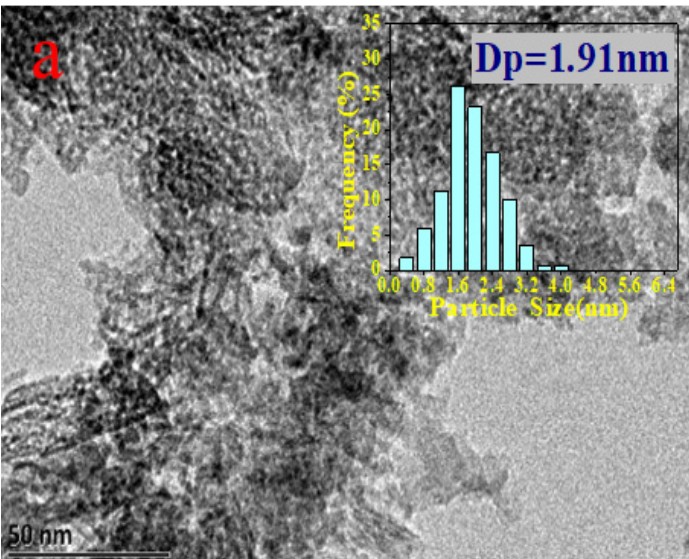

**Figure 3.** *Cont.*

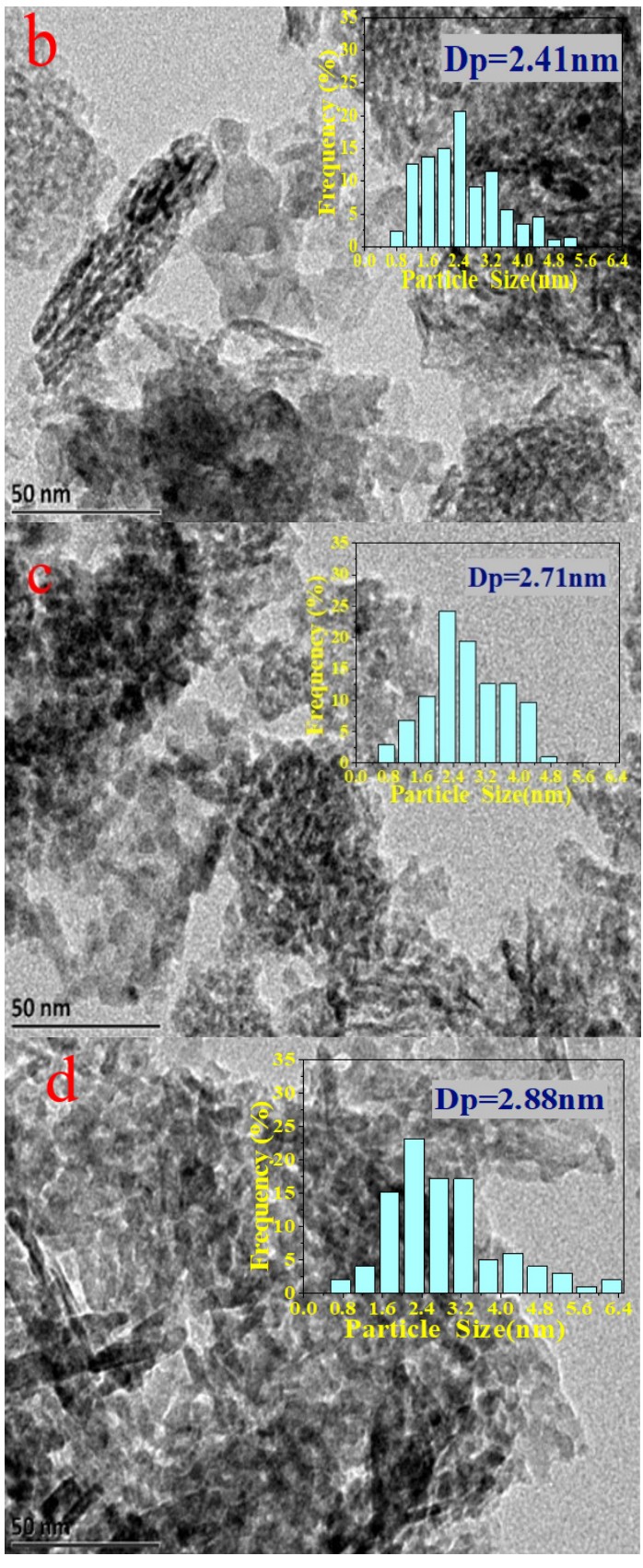

**Figure 3.** TEM images of Ni$_{10}$Mo$_{10}$Al catalysts calcined at (**a**) 450 °C, (**b**) 550 °C, (**c**) 650 °C and (**d**) 750 °C after reduction at 450 °C.

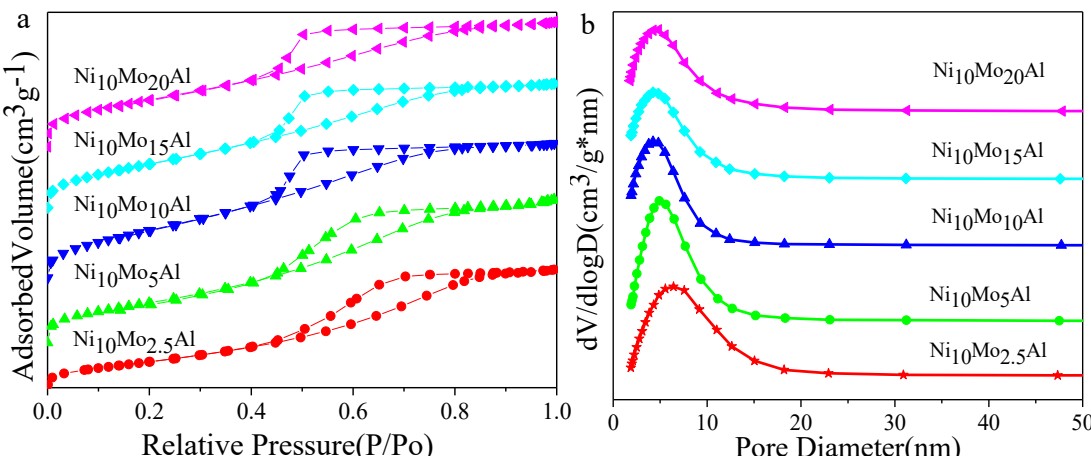

**Figure 4.** The $N_2$ adsorption–desorption isotherms (**a**) and the distributions of pore size (**b**) of $Ni_{10}Mo_xAl$ catalysts.

For the $Ni_{10}Mo_xAl$ catalyst, as presented in Table 2, the $N_2$-BET surface area ($S_{BET}$) showed little change with the increase in Mo content from 2.5 to 10 wt.% and then decreased with a further increase in $MoO_3$ content. This was due to the excess Mo content resulting in the agglomeration of the active components on the catalyst surface. The pore volume decreased from 0.48 to 0.29 $cm^3/g$, while the average pore diameter of the catalysts decreased from 4.81 to 3.17 nm.

**Table 2.** Texture properties of $Ni_{10}Mo_xAl$ catalysts.

| Catalyst | $S_{BET}$ [a] $(m^2/g)$ | Pore Volume [b] $(cm^3/g)$ | Average Pore Diameter [b] (nm) |
|---|---|---|---|
| $Ni_{10}Mo_{2.5}Al$ | 335 | 0.48 | 4.81 |
| $Ni_{10}Mo_5Al$ | 345 | 0.36 | 4.15 |
| $Ni_{10}Mo_{10}Al$ | 341 | 0.34 | 3.74 |
| $Ni_{10}Mo_{15}Al$ | 319 | 0.30 | 3.71 |
| $Ni_{10}Mo_{20}Al$ | 314 | 0.29 | 3.17 |

[a] Determined from $N_2$ physisorption using the BET method; [b] Determined from $N_2$ physisorption using the BJH method.

Figure 5a presents the XRD patterns of the $Ni_{10}Mo_xAl$ catalysts with different $MoO_3$ content. The diffraction peaks at $2\theta = 37.6$, 45.9 and 67.0° are in respect to the (311), (400) and (440) crystal plane of $\gamma–Al_2O_3$, respectively (JCPDS card no. 10-0425). The peaks at $2\theta = 37.2$, 43.3 and 62.9° are attributed to the (111), (200) and (220) crystal planes of NiO (JCPDS card no. 47-1049). The peaks at $2\theta = 19.1$, 37.0, 45.0 and 65.5° belong to the (111), (311), (400) and (440) crystal planes of $NiAl_2O_4$ (JCPDS card no. 10-0339) [28–30]. It was observed that the diffraction peaks of NiO and $NiAl_2O_4$ overlapped with the diffraction peak of $\gamma–Al_2O_3$. In addition, the intensity of the characteristic peaks at $2\theta = 37.6$, 45.9 and 67.0° gradually decreased with increasing $MoO_3$ content; this change may be due to the decrease in the $\gamma–Al_2O_3$ and $NiAl_2O_4$ phase content. It was noticed that no significant diffraction peak related to $MoO_3$ was observed, indicating that amorphous molybdenum oxide is formed on the catalyst surface in relation to the $MoO_3$ content. For the reduced $Ni_{10}Mo_xAl$ catalysts, as shown in Figure 5b, the characteristic peaks around $2\theta = 44.5$ and 51.8° corresponded to the (111) and (200) crystal planes of $Ni^0$ (JCPDS card no. 04-0850). The characteristic peak at $2\theta = 44.5°$ also overlapped with the diffraction peaks of the $NiAl_2O_4$ and $\gamma–Al_2O_3$ phases. However, the intensity of the characteristic peaks of $Ni^0$ at $2\theta = 51.8°$ gradually increased with the increase in $MoO_3$ content, indicating that the introduction of $MoO_3$ can increase the content of the reducible NiO which may be due to the decrease in the $NiAl_2O_4$ phase.

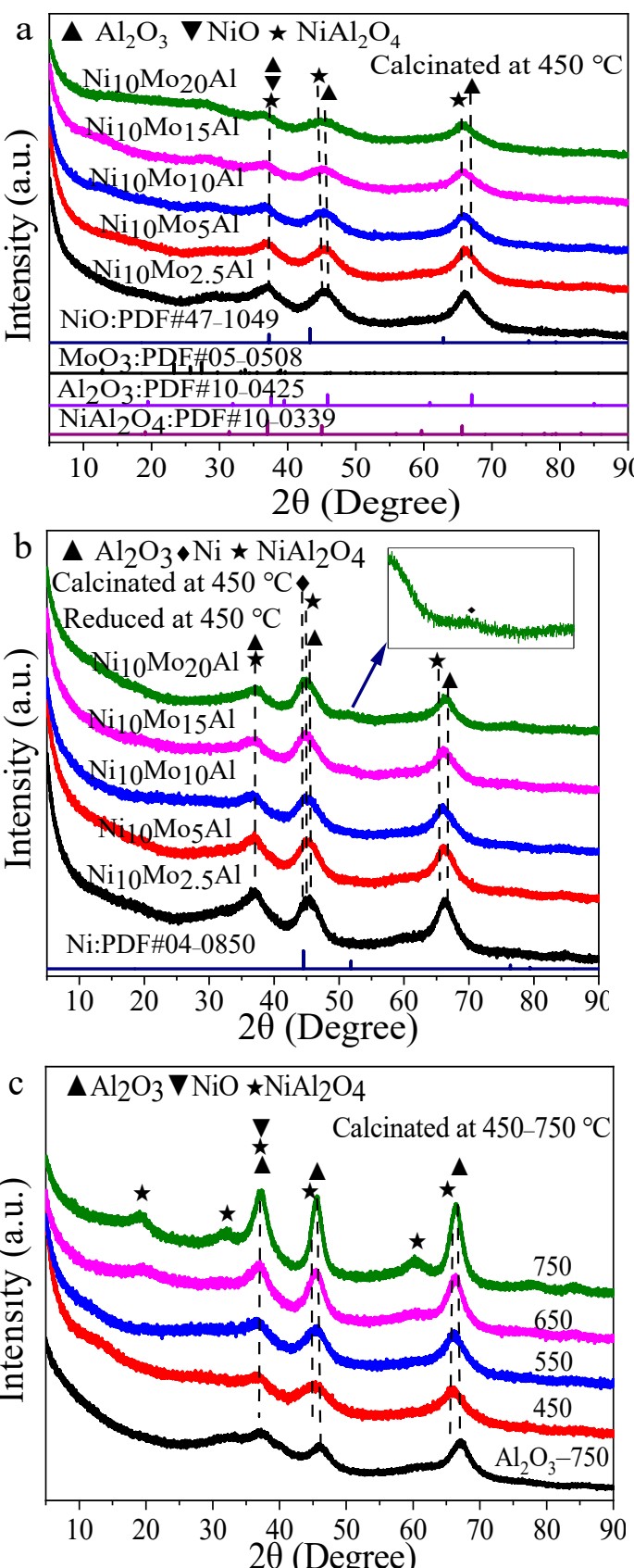

**Figure 5.** *Cont.*

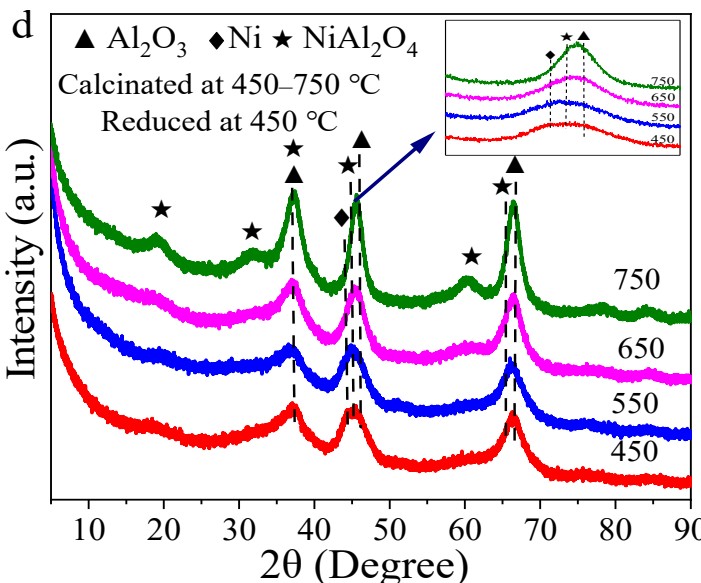

**Figure 5.** XRD patterns of $Ni_{10}Mo_xAl$ catalysts (**a**) before and (**b**) after reduction at 450 °C; $Ni_{10}Mo_{10}Al$ catalysts calcined at 450–750 °C (**c**) before and (**d**) after reduction at 450 °C.

Figure 5c shows the XRD spectra of $Ni_{10}Mo_{10}Al$ catalysts calcined at different temperatures (450–750 °C). With the increase in the calcination temperature, the diffraction peaks of the $\gamma$–$Al_2O_3$ and $NiAl_2O_4$ phases gradually became intense and narrow. The new diffraction peaks around $2\theta$ = 19.1, 31.4 and 60.9°, which were the overlapping peaks of $NiAl_2O_4$ and $Al_2O_3$, were observed when the catalyst was calcined under 750 °C. Compared with the diffraction peaks of $Al_2O_3$ calcinated at 750 °C, it can be concluded that the new diffraction peaks are mainly due to the formation of $NiAl_2O_4$ at a higher calcination temperature [31,32]. Regarding the reduced $Ni_{10}Mo_{10}Al$ catalysts, as shown in Figure 5d, the intensity of the characteristic peaks was attributed to $Ni^0$ decreases, while the peaks of the $NiAl_2O_4$ phase gradually became intense and narrow with the enhancement of the calcination temperature, demonstrating that the higher calcination temperature favors the formation of $NiAl_2O_4$.

Due to the overlap of the characteristic peaks between the $\gamma$–$Al_2O_3$ and $NiAl_2O_4$ phase in the XRD patterns, the UV-vis diffuse reflectance spectroscopic characterization was utilized to study the influence of the Mo element on the formation of the $NiAl_2O_4$ phase in the catalyst. For comparison, the pure $NiAl_2O_4$ compound was prepared and confirmed by XRD characterization (Figure S4). As shown in Figure 6, it was observed that the broad absorption band of the $NiAl_2O_4$ and NiO phases appeared around 580–670 nm and 670–765 nm, respectively [33,34]. It can be seen from Figure 6a that the adsorption band of the $NiAl_2O_4$ phase became weaker with the increase in the Mo content in the catalyst. Therefore, it can be concluded that the introduction of the Mo element can inhibit the formation of the $NiAl_2O_4$ phase. Regarding the effect of the calcination temperature, as shown in Figure 6b, the absorption band of the $NiAl_2O_4$ phase became intense with the increase in the calcination temperature, indicating that the high temperature leads to the formation of the $NiAl_2O_4$ phase. The catalyst color changed from gray to green also demonstrating this change. For comparison, the $Ni_{10}Mo_{10}/\gamma$–$Al_2O_3$ catalyst was also performed, as shown in Figure 6c. The band of the $Ni_{10}Mo_{10}Al$ catalysts was stronger than the $Ni_{10}Mo_{10}/\gamma$–$Al_2O_3$ catalysts, indicating that more $NiAl_2O_4$ phase was formed in the solvothermal preparation process.

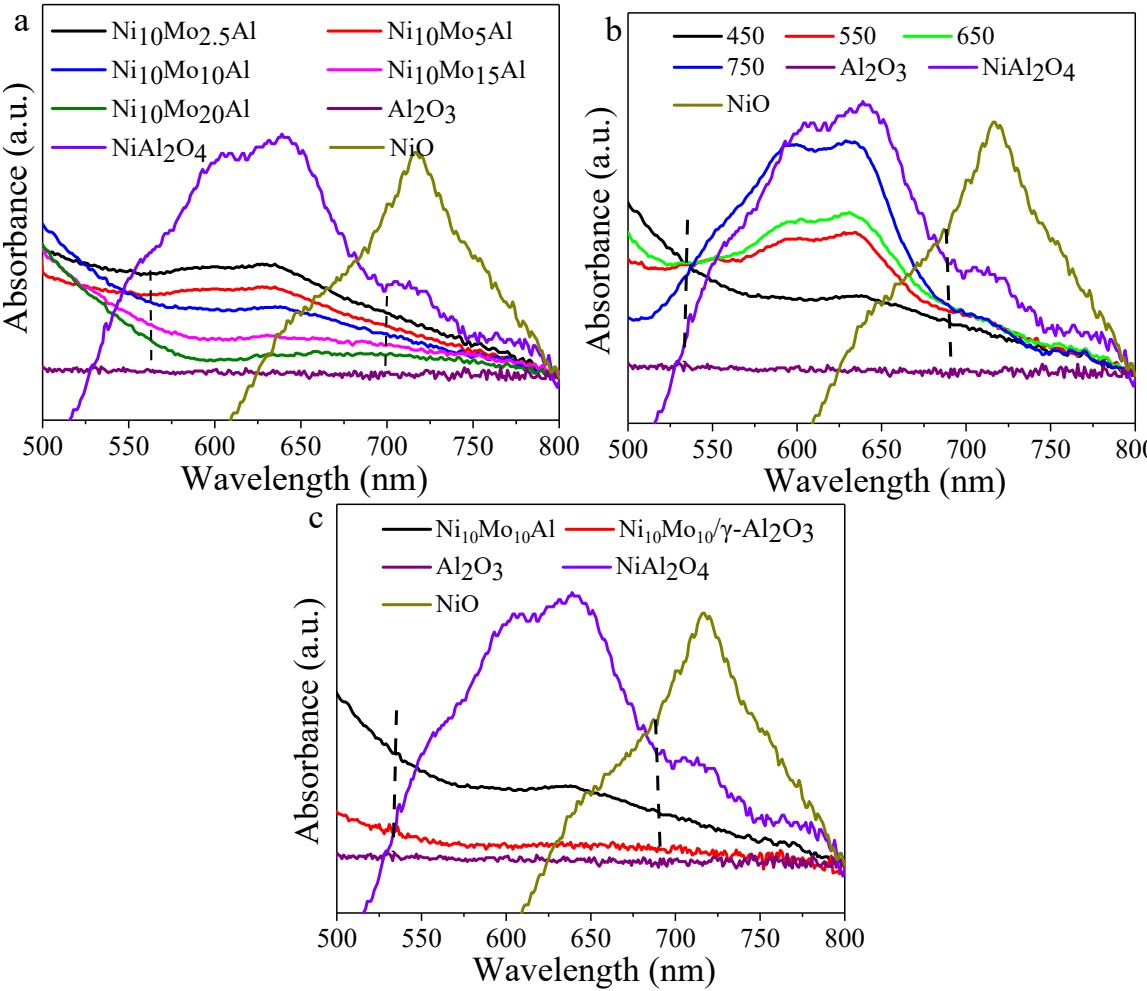

**Figure 6.** UV-vis DRS spectra of (**a**) $Ni_{10}Mo_xAl$ catalysts, (**b**) $Ni_{10}Mo_{10}Al$ catalysts calcined at 450–750 °C and (**c**) $Ni_{10}Mo_{10}Al$ and $Ni_{10}Mo_{10}/\gamma–Al_2O_3$ catalysts calcined at 450 °C.

The reduction properties of the catalysts were studied using $H_2$-TPR, and the results are shown in Figure 7. The reducibility of NiO depends on the interaction between NiO and $\gamma–Al_2O_3$. Regarding the $Ni_{10}Al$ catalyst, as shown in Figure S5, the $H_2$ consumption peak around 340 °C is assigned to the reduction in NiO. The reduction peak range around 485 °C and 600 °C is in respect to the weak and strong interaction between NiO and $\gamma–Al_2O_3$, respectively. In addition, the peak at a temperature above 700 °C is assigned to the reduction in the $NiAl_2O_4$ phase. Regarding the $Mo_{10}Al$ catalyst, as shown in Figure S5, the reduction peak at 210–400 °C was lower than the bulk molybdena, indicating that well-dispersed oxomolybdenum-containing species are easier to reduce than bulk molybdena [24,35]. The reduction peak at 450 °C is assigned to the reduction in $Mo^{6+}$ which interacted with the $\gamma–Al_2O_3$, and the peak at 660 °C is assigned to the reduction in $Mo^{4+}$. Regarding the $H_2$-TPR profiles of the $Ni_{10}Mo_xAl$ catalysts, the peak at 350 °C is corresponding to the NiO and $MoO_3$ in the free state. The peak at 450 °C is ascribed to the reduction in $Mo^{6+}$ and NiO, which showed a weak interaction with the $\gamma–Al_2O_3$ phase. The peak at 590 °C is assigned to the reducible NiO which showed a strong interaction with the $\gamma–Al_2O_3$ phase, while the hydrogen consumption peak at 750 °C is assigned to the overlap of the reduction peak of the $NiAl_2O_4$ phase and $Mo^{4+}$. It was noticed that the reduction peaks of NiO at 590 °C became intense with the increase in Mo content. Meanwhile, the peak of the reducible NiO shifted to a lower temperature with the increase in Mo content. These results suggest that the interaction between NiO and $Al_2O_3$ becomes weak [36].

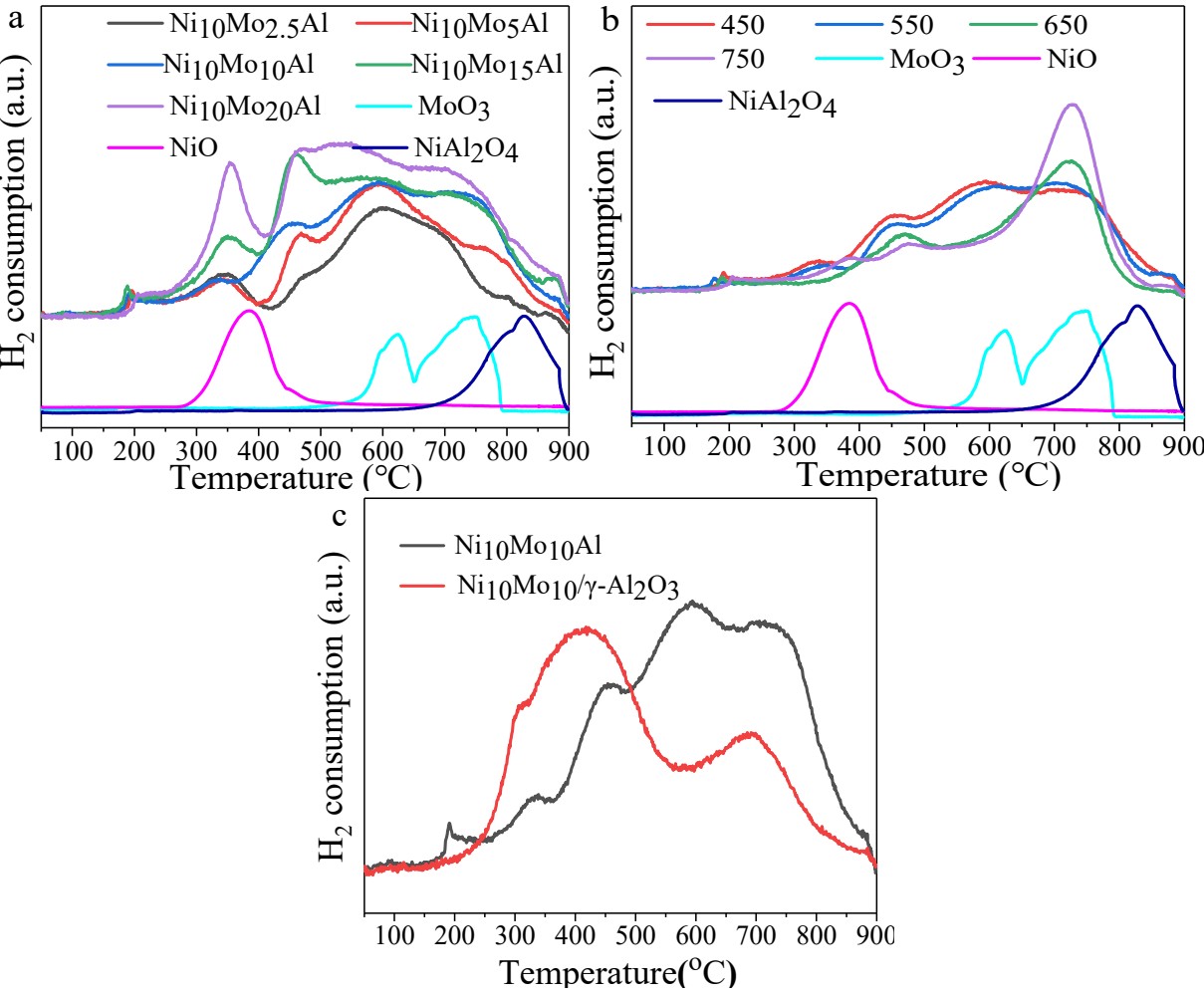

**Figure 7.** $H_2$-TPR profiles of (**a**) $Ni_{10}Mo_xAl$, (**b**) $Ni_{10}Mo_{10}Al$ catalysts calcined at 450–750 °C and (**c**) $Ni_{10}Mo_{10}Al$ and $Ni_{10}Mo_{10}/\gamma$–$Al_2O_3$ catalysts calcined at 450 °C.

The $H_2$-TPR profiles of $Ni_{10}Mo_{10}Al$ catalysts calcined at different temperatures are shown in Figure 7b. The $H_2$ consumption peaks (<600 °C) of reducible NiO and $MoO_x$ shifted to higher temperatures with the enhancement of the calcination temperature, indicating that the higher calcination temperature promotes the interaction between Ni–$Al_2O_3$ and Mo–$Al_2O_3$. In addition, the intensity of the reduction peaks (<600 °C) gradually decreased when the calcination temperature was raised from 450 to 750 °C. However, the reduction peak at 750 °C became gradually intense, indicating the formation of $NiAl_2O_4$ through the interaction between the Ni species and $Al_2O_3$ at a high calcination temperature, which is in agreement with the XRD and UV-vis characterization results.

The $H_2$-TPR profiles of the $Ni_{10}Mo_{10}/\gamma$–$Al_2O_3$ catalysts prepared using the incipient wetness method are shown in Figure 7c. It was noted that the reduction peaks regarding NiO and $MoO_3$ shifted towards a lower temperature than the $Ni_{10}Mo_{10}Al$ catalyst. The change demonstrated that the catalyst prepared using the solvothermal method showed a stronger interaction between Ni–Al and Mo–Al, which is conductive to promote the dispersion of the active components in the catalyst. In addition, the intensity of the $NiAl_2O_4$ reduction peak in the $Ni_{10}Mo_{10}Al$ catalyst was higher than that of the $Ni_{10}Mo_{10}/\gamma$–$Al_2O_3$ catalysts, indicating that more of the $NiAl_2O_4$ phase is in the $Ni_{10}Mo_{10}Al$ catalyst, which is in full agreement with the UV-vis diffuse reflectance spectroscopic characterization.

The reduced catalysts were measured by XPS analysis to investigate the chemical state of the Ni and Mo species on the catalyst surface, and their deconvolution are shown in Figure 8. All the results were corrected using the peak of C1s at 284.8 eV [37,38]. In

addition, the related parameters are listed in Table 3. As shown in Figure 8a, the binding energy around 852.5 eV is attributed to $Ni^0$, while 855.5 eV is assigned to $Ni^{2+}$ in NiO, and 857.5 eV is ascribed to $Ni^{2+}$ in $NiAl_2O_4$. The band at 862.2 eV is attributed to the satellite peak of $Ni2p_{3/2}$ [25,36]. With the increase in Mo content, as showed in Figure 8a, the binding energy of $Ni^0$ shifted to the higher binding energy, which may be related to the modification of the $Ni^0$ surface by $MoO_x$ [26]. According to the XPS characterization results, as shown in Table 3, the $Ni^0$ content on the catalyst surface rose, while the Ni–Al spinel content gradually decreased with the increase in Mo content, indicating that the introduction of the Mo element can inhibit the formation of the $NiAl_2O_4$ phase in the catalyst. This conclusion is also consistent with the UV-vis diffuse reflectance spectroscopic characterization results. Figure 8b depicts the Mo 3d spectra, and the quantitative analysis of the Mo species is listed in Table 3. The deconvolution results of the spectra show three different oxidation states around 228.8, 231.7 and 233.0 eV with respect to $Mo^{4+}$, $Mo^{5+}$ and $Mo^{6+}$, respectively [39,40]. It is obvious that the binding energy of Mo shifted to a lower binding energy, and the percentage of the $Mo^{6+}$ species increased with the increasing content of the Mo element.

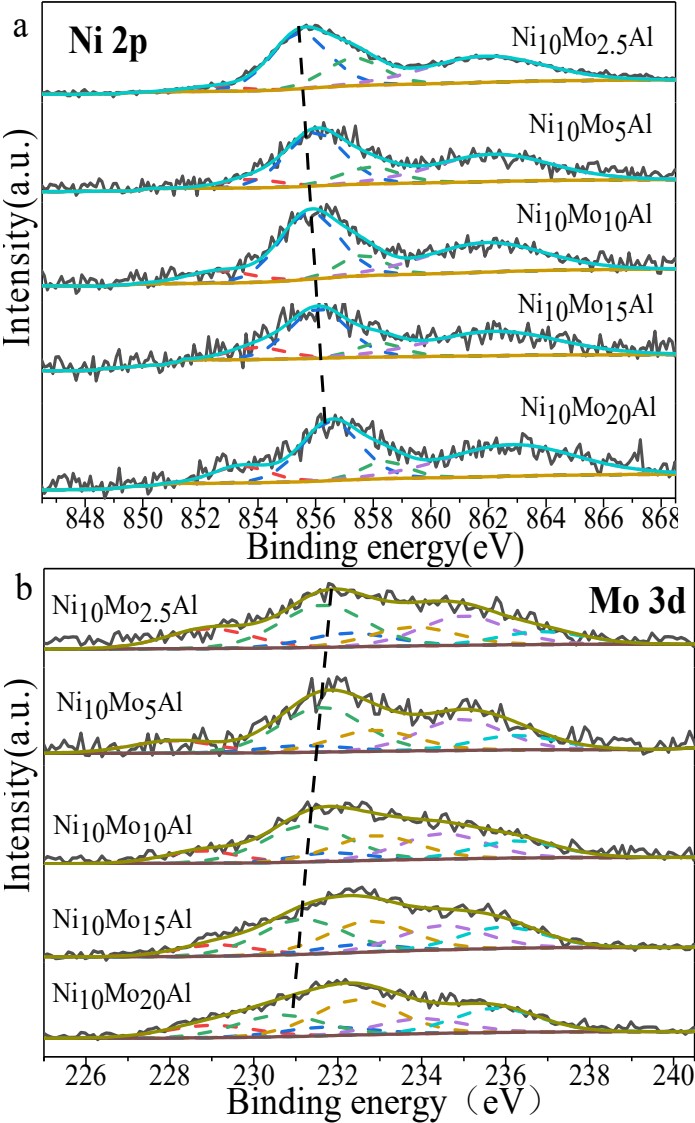

**Figure 8.** XPS spectra of Ni2p (**a**) and Mo3d (**b**) of $Ni_{10}Mo_xAl$ catalysts reduced at 450 °C.

**Table 3.** Quantitative Ni 2p and Mo 3d XPS analysis for $Ni_{10}Mo_xAl$ catalysts [a].

| Sample | Percentage of Surface Mo and Ni Species Detected by XPS (%) | | | | | |
|---|---|---|---|---|---|---|
| | $Mo^{6+}$ | $Mo^{5+}$ | $Mo^{4+}$ | $Ni^0$ | NiO | $NiAl_2O_4$ |
| $Ni_{10}Mo_{2.5}Al$ | 17.2 | 46.3 | 37.5 | 5.5 | 68.6 | 25.9 |
| $Ni_{10}Mo_5Al$ | 21.2 | 44.8 | 34.0 | 10.9 | 67.9 | 21.2 |
| $Ni_{10}Mo_{10}Al$ | 27.6 | 41.0 | 31.4 | 14.8 | 65.7 | 19.5 |
| $Ni_{10}Mo_{15}Al$ | 33.3 | 37.8 | 28.9 | 17.7 | 64.4 | 17.9 |
| $Ni_{10}Mo_{20}Al$ | 45.4 | 28.1 | 26.5 | 21.3 | 62.9 | 15.8 |

[a] All the values are based on the peak area percentage.

Figure 9 presents the Ni 2p and Mo 3d spectra of the reduced (at 450 °C) $Ni_{10}Mo_{10}Al$ catalyst after calcination at different temperature (450–750 °C), and the related analysis results are listed in Table 4. As shown in Figure 9a, the peak intensity of the $Ni^0$ species gradually decreased with the increasing calcination temperature, while it gradually increased for the $NiAl_2O_4$, which is consistent with the $H_2$-TPR characterization results. In addition, the increasing content of $Mo^{6+}$ indicates that a higher calcination temperature promotes the interaction between Mo–Al.

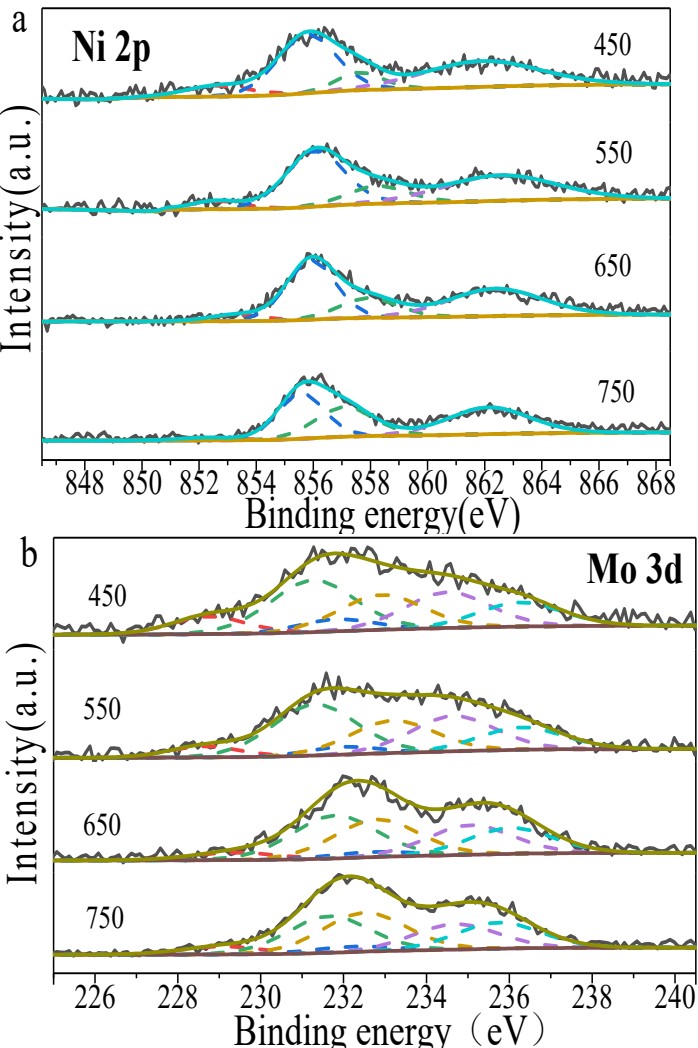

**Figure 9.** XPS spectra of (**a**) Ni 2p and (**b**) Mo 3d of $Ni_{10}Mo_{10}Al$ catalysts after calcination at 450–750 °C.

**Table 4.** Quantitative Ni 2p and Mo 3d XPS analysis for reduced $Ni_{10}Mo_{10}Al$ catalysts after calcination at 450–750 °C [a].

| Sample | Percentage of Surface Mo and Ni Species Detected by XPS (%) | | | | | |
|---|---|---|---|---|---|---|
| | $Mo^{6+}$ | $Mo^{5+}$ | $Mo^{4+}$ | $Ni^0$ | NiO | $NiAl_2O_4$ |
| O-450 | 27.6 | 41.0 | 31.4 | 14.8 | 65.7 | 19.5 |
| O-550 | 31.7 | 39.8 | 28.5 | 11.0 | 63.3 | 25.7 |
| O-650 | 36.8 | 37.0 | 26.2 | 9.2 | 61.4 | 29.4 |
| O-750 | 38.6 | 36.0 | 25.4 | 3.8 | 52.8 | 43.4 |

[a] All the values are based on the peak area percentage.

## 2.2. Catalytic Evaluation

The effect of the Mo content on catalytic hydrogenation activity is shown in Figure 10. It was found that no byproducts were detected during the hydrogenation of MA under mild reaction conditions, which was consistent with the previous study [7–10]. So, the selectivity of MP remained stable at 100% in the wide range of Mo content from 2.5 to 20 wt.%. However, the yield of MP showed a volcanic pattern, and the $Ni_{10}Mo_{10}Al$ catalyst existed at the peak position. According to the above TEM characterization results, the particle size of the Ni species on these catalyst series followed in the order of $Ni_{10}Mo_{10}Al < Ni_{10}Mo_5Al < Ni_{10}Mo_{2.5}Al < Ni_{10}Mo_{15}Al < Ni_{10}Mo_{20}Al$, which is consistent with the difference in the catalytic activity of $Ni_{10}Mo_{2.5}Al$, $Ni_{10}Mo_5Al$ and $Ni_{10}Mo_{10}Al$. As for the $Ni_{10}Mo_{15}Al$ and $Ni_{10}Mo_{20}Al$ catalysts, their catalytic activities were still both higher than that of $Ni_{10}Mo_{2.5}Al$, although the particle size of the Ni species in them was larger, considering the lower content of the $NiAl_2O_4$ phase. In conclusion, the particle size of the Ni species and the content of the $NiAl_2O_4$ phase in this kind of $Ni_{10}Mo_xAl$ composite oxide catalyst are both crucial to the catalytic hydrogenation activity. This experimental phenomenon also demonstrates that the appropriate content of the Mo element not only promotes the dispersion of the Ni species but also decreases the formation of the $NiAl_2O_4$ phase; however, the excessive Mo covers the Ni particles and block the pore channels, which is consistent with the SEM and TEM characterization results [30,41].

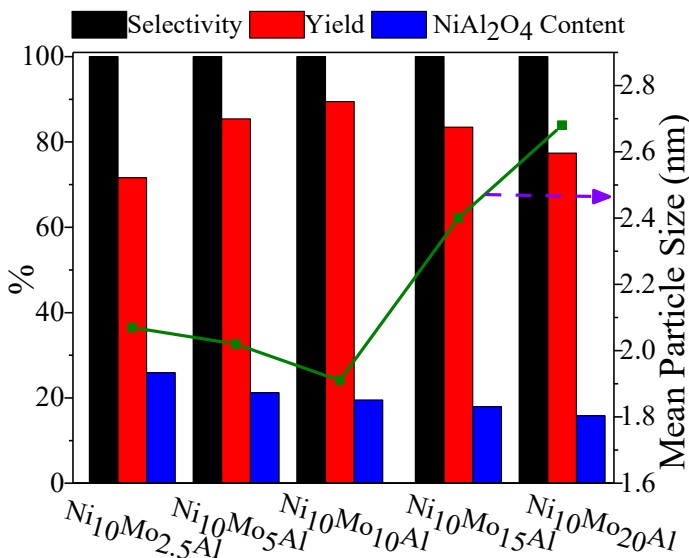

**Figure 10.** Catalytic evaluation of $Ni_{10}Mo_xAl$ catalyst series at the reaction conditions of 100 °C, 1 MPa $H_2$, $n(H_2)/n(MA) = 4$ and WHSV = 2 h$^{-1}$.

For comparison, the catalytic activity of $Ni_{10}Mo_{10}/\gamma–Al_2O_3$ prepared using a traditional impregnation method was also evaluated under the same reaction conditions, and the yield of MP was only 53.7%, with a selectivity of 100%, as shown in Table S1. This is

mainly due to the small particle size and better distribution of the active components in the catalyst prepared using the solvothermal method, although the solvothermal method tends to generate Ni–Al spinel, as demonstrated by the TEM and UV-vis characterization results.

First, the effect of a reduction in temperature was studied by reducing the $Ni_{10}Mo_{10}Al$ catalyst to 300–500 °C, and the hydrogenation result is presented in Figure S5. It was observed that the highest yield of MP, 89.6%, was achieved at 450 °C. So, the catalytic activities of the $Ni_{10}Mo_{10}Al$ catalysts calcinated at 450–750 °C and reduced at 450 °C were evaluated for the investigation of the effect of the calcination temperature, and the result is shown in Figure 11. With an increase in the calcination temperature, both the particle size of the Ni species and the content of the $NiAl_2O_4$ phase increased, and as a result, the catalytic hydrogenation activities on these $Ni_{10}Mo_{10}Al$ catalysts decreased accordingly. This result also suggests that the high dispersion of the Ni species and the inhibition of $NiAl_2O_4$ phase in such $Ni_{10}Mo_{10}Al$ catalysts promotes hydrogenation activity.

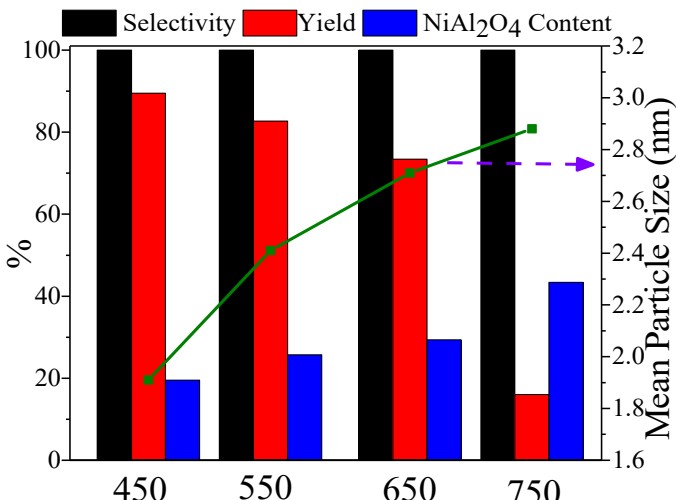

**Figure 11.** Effect of calcination temperature on the catalytic activity of $Ni_{10}Mo_{10}Al$ catalyst under the reaction conditions of 100 °C, 1 MPa $H_2$, $n(H_2)/n(MA) = 4$ and WHSV = 2 $h^{-1}$.

The effects of reaction conditions including temperature and pressure on MA hydrogenation were investigated, and the results are shown in Figure 12. As shown in Figure 12a, the selectivity of MP remained at 100%, while the yield increased from 68.5% to 89.6% when the reaction temperature was increased from 40 to 100 °C. However, both the yield and selectivity of MP tended to decrease when the temperature was above 100 °C due to the polymerization of MA. The influence of reaction pressure on the hydrogenation of MA is presented in Figure 12b. The selectivity of MP was stable at 100% with the increase in reaction pressure from 0.4 to 1.2 MPa, while the yield of MP increased from 68.1% to 89.6% and then remained unchanged with further increases. Therefore, the optimal reaction temperature is 100 °C, and the suitable reaction pressure of $H_2$ is 1.0 MPa.

The catalytic stability of $Ni_{10}Mo_{10}Al$ was tested under the optimal reaction conditions of 100 °C and 10 MPa, which is displayed in Figure 13. The selectivity of MP was still relatively stable at 100%; however, the yield of MP tended to decrease after 23 h time-on-stream. Comparing the ICP–OES analysis results of the fresh and deactivated catalysts, as listed in Table S2, it was observed that the Ni and Mo content reduced from 6.58 to 5.71 wt.% and from 6.07 to 5.81 wt.%, respectively. Moreover, the deactivated $Ni_{10}Mo_{10}Al$ catalyst was characterized using TEM, as shown in Figure S6. The particle size of the Ni species in the deactivated catalyst was larger than that in the fresh catalyst. Therefore, the deactivation of the catalyst can be attributed to the loss of Ni and Mo elements and the agglomeration of active components during the hydrogenation reaction.

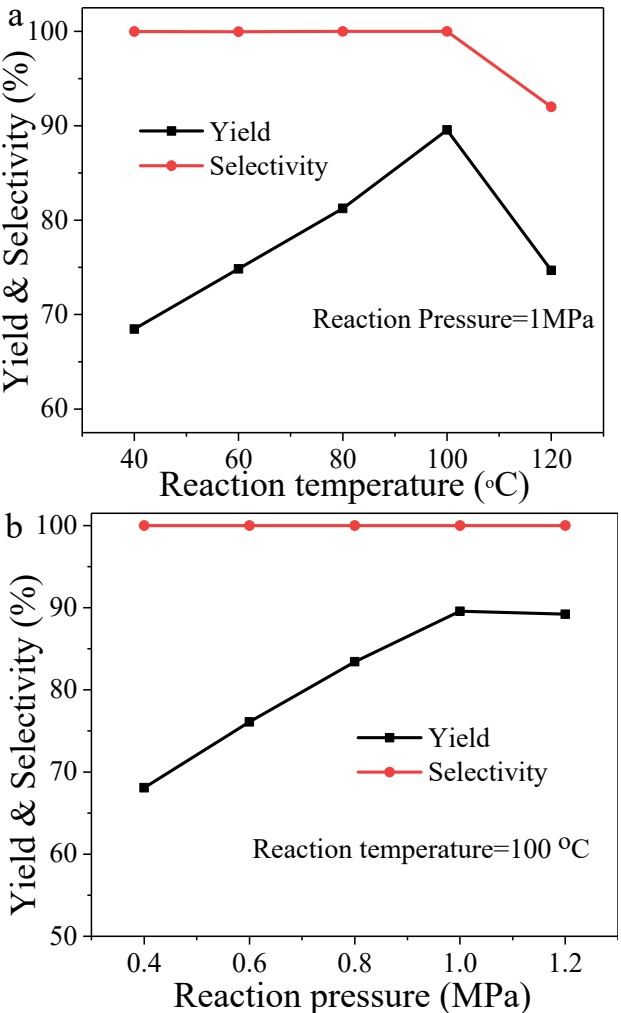

**Figure 12.** Effect of (**a**) reaction temperature and (**b**) pressure on the yield and selectivity of MP.

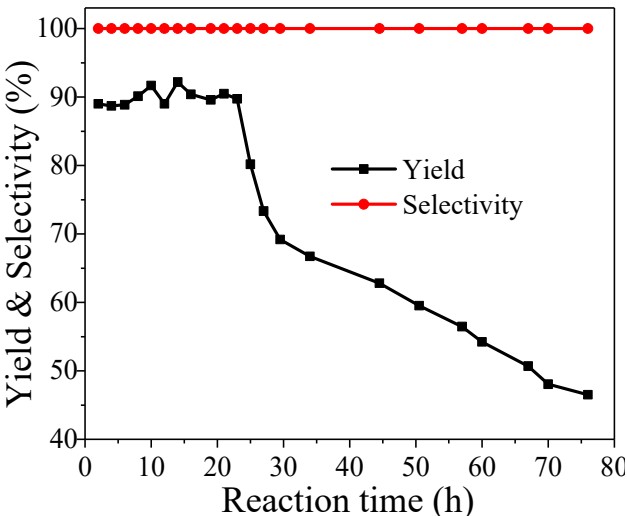

**Figure 13.** Catalytic stability test for $Ni_{10}Mo_{10}Al$ catalyst at the reaction conditions of 100 °C, 1 MPa $H_2$, $n(H_2)/n(MA) = 4$ and WHSV = $2 \, h^{-1}$.

## 3. Experimental Section

### 3.1. Catalyst Preparation

MA ($\geq$99.0%) and absolute ethyl alcohol ($\geq$99.7%) were offered by Damao Chemical Reagent Factory (Tianjin, China). $NiC_4H_6O_4\cdot4H_2O$ ($\geq$99.0%), $Na_2CO_3$ ($\geq$99.5%) and $Al(NO_3)_3\cdot9H_2O$ ($\geq$99.0%) were provided by Macklin Biochemical Co., Ltd. (Shanghai, China). NaOH ($\geq$99.5%) was offered by Xilong Co., Ltd. (Shantou, China). n-Hexane ($\geq$99.0%) and $(NH_4)_6Mo_7O_{24}\cdot4H_2O$ ($\geq$99.0%) were supported by the Sinopharm Chemical Reagent Co., Ltd. (Shanghai, China). All reagents were used without any further purification.

$Ni_{10}Mo_xAl$ catalysts were prepared using a solvothermal method (Figure 14). First, solution A was prepared by adding the required amount of $NiC_4H_6O_4\cdot4H_2O$ and $Al(NO_3)_3\cdot9H_2O$ in 50 mL ethanol. Solution B was prepared by dissolving the desired amount of $(NH_4)_6Mo_7O_{24}\cdot4H_2O$, $Na_2CO_3$ and NaOH in 50 mL deionized water. Then the solution B was introduced dropwise in solution A under vigorous stirring for 30 min, during which the pH was controlled between 7–8. Then the mixture was transferred into 150 mL Teflon-lined autoclave and placed at 110 °C for 16 h. The obtained powders were washed with water and ethanol several times. Then the sample was dried at 60 °C for 8 h and calcined at 450 °C for 4 h to obtain $Ni_{10}Mo_xAl$. For comparison, the $Ni_{10}Al$ and $Mo_{10}Al$ catalysts used as reference samples were prepared in the same conditions. The $Ni_{10}Mo_{10}/\gamma-Al_2O_3$ catalyst and $NiAl_2O_4$ were, respectively, prepared by the incipient impregnation and coprecipitation method, which has been reported in our previous work [18]. Before the hydrogenation experiments, the catalyst was reduced in situ at 450 °C for 4 h with a continuous flow of 10% $H_2/N_2$.

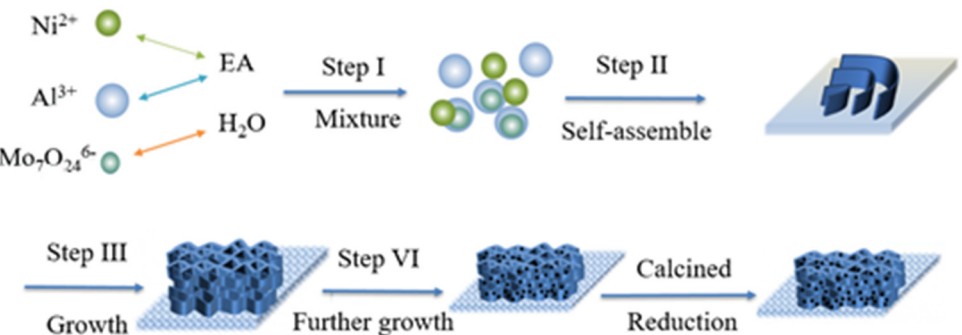

**Figure 14.** Schematic illustrating for the preparation of $Ni_{10}Mo_xAl$ catalyst.

### 3.2. Catalyst Characterization

Scanning electron microscope (SEM) images of the catalysts were obtained on a Hitachi SUB8020 instrument. Transmission electron microscope (TEM) and high resolution transmission electron microscope (HRTEM) examination were performed on a FEI Talos F2000x instrument. Powder X-ray diffraction patterns (XRD) were examined on a Rigaku Smart Lab X-ray powder diffractometer with Cu K$\alpha$ radiation. The diffraction patterns were recorded from 2$\theta$ = 5° to 90° under a 10°/min scanning speed. UV-vis diffusion reflectance spectroscopy (UV-vis DRS) was recorded on a UV-2550 spectrophotometer, and the diffractograms were carried out in the wavelength of 500–800 nm, with $BaSO_4$ powder as background. $N_2$ adsorption–desorption isotherms of the catalysts were performed on a micromeritics ASAP 2460 apparatus. The surface area was calculated using the Brunauer–Emmett–Teller (BET) method, and the pore size was obtained using the Barrett–Joyner–Halenda (BJH) model. The content of the Ni and Mo elements in the catalyst were analyzed by ICP-OES on a PerkinElmer Optima 8000 instrument. Before testing, the catalyst powder was ground to 200 mesh and dissolved in HF solution. Then the mixture was purified with a disposable syringe and appropriately diluted. Hydrogen temperature programmed reduction ($H_2$-TPR) of the catalysts was carried out on the AutochemII 2920 chemisorption apparatus (Micromeritics, Norcross, GA, USA). The sample was placed in a U-shaped quartz reactor and treated under flowing He at 300 °C for 1 h to remove the

adsorbed water and impurities and then cooled to 50 °C. Subsequently, the 10% $H_2$/Ar gas was introduced to the sample, and the temperature was raised to 900 °C with a heating rate of 5 °C/min. X-ray photoelectron spectroscopy (XPS) analysis was obtained on an Escalab 250Xi electron spectrometer using Mg X-ray source. All the binding energies of the catalysts were corrected with C 1s (284.8 eV) as the reference.

*3.3. Catalytic Testing*

The hydrogenation of MA was carried out in a fixed bed reactor with a vertical tube (d = 8 mm). First, the catalyst samples were pretreated in 10% $H_2$ flow at 450 °C for 4 h, and then the temperature was decreased to the reaction temperature. After that, the reactant of MA was introduced into the reactor using a constant flux pump under reaction conditions. The analysis of the products was performed on GC (SP-7890) equipped with an OV1701 column and a hydrogen flame detector, using n-hexane as internal standard substance. The catalytic activity was evaluated using MA conversion and MP yield and selectivity, which were calculated as Equations (1)–(3).

$$\text{Conversion of MA} = \frac{[MA]\text{in, mol} - [MA]\text{out, mol}}{[MA]\text{in, mol}} \times 100\% \tag{1}$$

$$\text{Selectivity of MP} = \frac{[MP]\text{out, mol}}{[MA]\text{in, mol} - [MA]\text{out, mol}} \times 100\% \tag{2}$$

$$\text{Yield of MP} = \text{Conversion of MA} \times \text{selectivity of MP} \tag{3}$$

## 4. Conclusions

A series of $Ni_{10}Mo_xAl$ catalysts were prepared using the solvothermal method in a water–ethanol system due to the characteristics that Ni ions precipitate in alkaline environments and Mo ions precipitate in acidic conditions, which also provides a new method for metal catalysts with opposite precipitation pH values. The introduction of Mo can improve the dispersion of the Ni species and inhibit the formation of Ni–Al spinel, which significantly affect the catalytic activity in the MA hydrogenation to MP. As a result, the $Ni_{10}Mo_{10}Al$ catalyst exhibits 89.6% yield of MP with 100% selectivity under the optimized conditions of 100 °C and 1 MPa, which is higher than the 53.7% yield achieved by $Ni_{10}Mo_{10}/\gamma–Al_2O_3$ prepared using a traditional impregnation method. However, the deactivation behavior observed on this kind of $Ni_{10}Mo_{10}Al$ composite catalyst is derived from the loss of Ni and Mo elements and the agglomeration of the Ni species. It is foreseeable that if the stability of the catalyst is improved, the catalyst prepared by the new strategy will have broader industrial application potential.

**Supplementary Materials:** The following supporting information can be downloaded at: https://www.mdpi.com/article/10.3390/catal12101118/s1, Figure S1. SEM images of (a) $Mo_{10}Al$, (b) Ni10Al catalysts; Figure S2. TEM images of reduced $Ni_{10}Mo_{10}/\gamma–Al_2O_3$ catalyst after reduction at 450 °C; Figure S3. XRD patterns of $NiAl_2O_4$; Figure S4. $H_2$-TPR profiles of $Ni_{10}Al$, $Mo_{10}Al$, $Ni_{10}Mo_{10}Al$ catalysts; Figure S5. Effect of reduction temperature on the catalytic activity of $Ni_{10}Mo_{10}Al$ catalyst at the reaction conditions of 100 °C, 1 MPa $H_2$, n(H2)/n(MA) = 4 and WHSV =2 $h^{-1}$; Figure S6. TEM images of the used $Ni_{10}Mo_{10}Al$ catalyst; Table S1. Properties of $NiAl_2O_4$, $Ni_{10}Mo_{10}Al$ and $Ni_{10}Mo_{10}/\gamma–Al_2O_3$ catalysts; Table S2. ICP–OES results for the fresh and used $Ni_{10}Mo_{10}Al$ catalyst.

**Author Contributions:** Conceptualization, Z.L. and E.W.; formal analysis, T.S.; formal analysis, T.S.; data curation, T.S.; Original draft of the paper, T.S.; Review, G.W. and X.G.; writing—review and editing, T.S. and Z.L. Supervision and funding: Z.L. and C.L. All authors have read and agreed to the published version of the manuscript.

**Funding:** This research was funded by the National Natural Science Fund for Distinguished Young Scholars (Grant number: 22025803), the National Natural Science Foundation of China (Grant number: 22178338), the Innovation Academy for Green Manufacture, Chinese Academy of Sciences (Grant number: IAGM-2019-A14) and the Key R&D Plans of Hebei Province (Grant number: 20374002D).

**Data Availability Statement:** Not applicable.

**Conflicts of Interest:** The authors declare no conflict of interest.

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
