# Peer review of "A Highly Active NiMoAl Catalyst Prepared by a Solvothermal Method for the Hydrogenation of Methyl Acrylate"

_catalysts, doi:10.3390/catal12101118_

Round 1

Reviewer 1 Report

In the presented manuscript the authors present the preparation of Ni10MoxAl composite metal oxide (Ni10MoxAl) catalysts with different Mo content (x= 2.5, 5, 10, 15, 20 wt.%)which were prepared by a solvothermal method using a water−ethanol system. Further, all the catalysts have been characterized by   X−ray diffraction (XRD), Brunauer−Emmett−Teller analysis (BET), UV−vis diffuse reflectance spectroscopy (UV−vis DRS), and hydrogen temperature programmed reduction (H2−TPR), X−ray photoelectron spectroscopy (XPS), and transmission electron microscopy (TEM).

The presented research is interesting and detailed. The results are well structured and nicely shaped within the presentation.

The catalytic evaluation was investigated through several parameters: molybdenum content and the calcination temperature.

I would suggest broadening the conclusion part for future research and perspectives. The obtained results gave insight into the obtained systems, but are potentially opening a new niche of the research that could be discussed.

For this reason, I suggest minor revisions for this manuscript.

Reviewer 2 Report

Reviewer comments on Manuscript 1859164

Title:

 A Highly Active NiMoAl Catalyst Prepared by Solvothermal Method for the Hydrogenation of Methyl Acrylate

Abstract:

Introduction: Literature on non-precious metals catalysts are missing; Please add literature summary  on non-precious metals from references 11-12 and also other than NiMo catalysts  and then establish the need of the current research and why you chose Ni10MoxAl  catalysts.  

Methods: All good

Results and discussion: The stability of the catalysts decrease after 24 hr and you added the reason. But you should mention the decrease in the stability in the abstract . Rest of the results and discussion is  very adequate.

Reviewer 3 Report

The method approaches and catalytic performance for the hydrogenation of MA are addressed in this manuscript, but some questions in this manuscript have to be solved. The manuscript needs revision and clarification.

1.      The authors mention that the hydrogenation activity of Ni10MoxAl catalysts was affected by the particle size of active components, how to control it? How about the optimum condition process to reach it?

2.      In Figure 1 after step VI, the word calicined should be calcined.

3.      Could the authors add the SEM result compared to the other previous research as mentioned in the introduction?

4.      The surface area, SBET changed little with the increase of Mo content, then decreased with a further rise in MoO3 content, due to the excess MoO3 content blocking the micropores. How the authors justified this statement, refer in table 2.

5.      Assigned in the catalytic evaluation, the author evaluated at the same reaction conditions and the yield of MP with the selectivity of MA relative constant of 100%, but in equation 2 mention the selectivity of MP, is that correct?

When the temperature is above 100oC, how to know the polymerization of MA? Please support it with the data/reaction mechanism.

Round 2

Reviewer 2 Report

Additional Comments and Suggestions for Authors: Reviewer comments on revised Manuscript 1859164

Title:

A Highly Active NiMoAl Catalyst Prepared by Solvothermal Method for the Hydrogenation of Methyl Acrylate

1.     In my original comments, I requested the authors to add references of non-precious metal catalysts other than NiMOAl . I don’t see any additional references on this.

2.     This is the authors reply to  my original comments: “According to our previous work, the Ni−Mo/γ−Al2O3 catalyst, prepared by incipient-wetness impregnation method, exhibited excellent activity in MA hydrogenation to MP at mild condition, however, it still needs to be improved for the industrial application”.

In support of this claim , it will be useful if the authors clarify under the Results/Discussion section of the second revised manuscript the following.

What are the new conditions and why it improved the reaction . Please add some details what are new in the new method. How the new method will be better useful for industrial application?

33. The author added the catalyst was stable for 23 hr. For industrial application, the catalyst stability must be for at least 72 hours. Hence, I suggest the authors do additional experiments on stability and add the new data to the second revision of the manuscript.        
